# A Novel Simulation Method of Micro-Topography for Grinding Surface

**DOI:** 10.3390/ma14185128

**Published:** 2021-09-07

**Authors:** Qi An, Shuangfu Suo, Yuzhu Bai

**Affiliations:** Department of Mechanical Engineering, Tsinghua University, Beijing 100084, China; sfsuo@tsinghua.edu.cn (S.S.); baiyuzhu403@163.com (Y.B.)

**Keywords:** grinding surface, microtopography, frequencies extraction, simulation method

## Abstract

A novel simulation method of microtopography for grinding surface was proposed in this paper. Based on the theory of wavelet analysis, multiscale decomposition of the measured topography was conducted. The topography was divided into high frequency band (HFB), theoretical frequency band (TFB), and low frequency band (LFB) by wavelet energy method. The high-frequency and the low-frequency topography were extracted to obtain the digital combination model. Combined with the digital combination model and the theoretical topography obtained by geometric simulation method, the simulation topography of grinding surface can be generated. Moreover, the roughness parameters of the measured topography and the simulation topography under different machining parameters were compared. The maximum relative error of *Sa*, *Sq*, *Ssk* and *Sku* were 1.79%, 2.24%, 4.69% and 4.73%, respectively, which verifies the feasibility and accuracy of the presented method.

## 1. Introduction

The simulation of machined surface was always an important topic in the field of tribology [1]. From the micro perspective, the surface topography contains abundant machining information [2]. If the surface topography can be reconstructed by means of simulation method, it is of great significance for predicting the surface topography and studying the contact behavior of rough surfaces [3,4,5].

Modeling methods of grinding surface topography were various. The numerical simulation method (NSM) and the geometric simulation method (GSM) were two commonly used simulation methods. The NSM simulated the ground surface by generating Gaussian or non-Gaussian surfaces [6]. Previously surface analysis was restricted by technical limitations. A digital characteristic of modeled profile was needed. Therefore, parameters describing profiles completely must be selected [1]. Wu simulated the grinding rough surfaces by generating Gauss and non-Gauss surfaces based on Fourier transform [7,8]. Reizer simulated the surface topography by 3D Gaussian surface [9]. Wang combined Fourier transform with Johnson conversion system [10] to simulate non-Gaussian surfaces with specific roughness parameters [11]. The simulation surface obtained by NSM was based on the roughness parameters, which can meet the statistical characteristics [1]. However, researchers found that the truncation length of autocorrelation function has a great influence on the simulation topography. Improper truncation length of autocorrelation function may lead to wrong results [12,13]. Moreover, using Gaussian or non-Gaussian surfaces instead of actual grinding surfaces will introduce errors into the analysis of the contact characteristics of grinding surfaces [4,14,15].

The GSM was based on the theory of grinding kinematics. Combined with the grinding parameters and the movement track of grinding wheel particles, the simulation of grinding surface topography was realized. Based on the grinding parameters, Warneck deduced the motion equation between the grinding wheel and the workpiece [16]. Based on the research of Warneck, Cooper added the empirical formula of plough and sliding friction in the process of topography simulation [17]. Nguyen developed a numerical simulation program for grinding surface topography based on grinding wheel topography data and grinding kinematics theory [18,19]. Based on the theory of Nguyen, Cao considered the influence of the vibration between the grinding wheel and the workpiece during the simulation of the surface topography [20,21]. Combined with Johnson transform [10], Wen conducted a simulation study on the theoretical grinding topography based on the topography of the grinding wheel and the grinding kinematics [22]. Chen introduced the surface waviness information into the simulation process of surface topography [23]. Lipiński applied artificial neural networks to the modelling of surface roughness and grinding forces [24]. The simulation surface obtained by GSM was based on the theory of grinding kinematics, which can get the corresponding rules of machining parameters and surface topography [1]. However, there were many influencing factors in the grinding process, and it was difficult to accurately and quantitatively detect each influencing factor [13]. Therefore, with the depth of research, the analysis difficulty of GSM will be greatly increased.

With the development of measurement technology, the analysis method based on measured surface became a new research topic [25]. From the perspective of signal decomposition, different machining factors correspond to different topography components. Then, researchers proposed a simulation method based on the measured topography. Zhao compared Haar, dbN, bior, symN, and other wavelet basis functions from the perspective of the overall error of reconstructed topography [26]. Pour applied the time series analysis and wavelet transform to determining surface roughness of the ground workpieces [27]. Goiec compared wavelet analysis and Gaussian filter in the constructing process of engineering surfaces, and confirmed the advantages of wavelet analysis [28]. Based on the above research, a novel simulation method based on wavelet analysis and GSM was proposed in this paper.

## 2. Simulation Method of Surface Topography

In this section, the causes of grinding surface topography were analyzed. Combined with the theory of wavelet analysis, the influencing factors in grinding process were classified and processed according to the frequency information. Ultimately, the simulation method of surface topography was proposed.

### 2.1. Cause Analysis of Surface Topography

According to different frequency information, the grinding surface topography can be divided into four components: roughness, waviness, geometric shape, and other random factors [29]. Combined with Figure 1, the influencing factors of each component were summarized as follows: (1) roughness was mainly caused by the machining parameters, grinding wheel parameters, and the high frequency vibration of the equipment; (2) waviness was mainly caused by the low-frequency vibration between the grinding wheel and the workpiece; (3) geometric shape was mainly caused by the feed error of grinder; (4) other random components were mainly caused by the thermal deformation of process system, as well as the surface defects of workpiece and the lubrication conditions.

Figure 1 demonstrates how there are numerous factors that can affect the formation of grinding surface topography. The GSM needs to analyze the various factors in the grinding process separately. The feasibility of this method was poor; therefore, the analysis will be carried out from the measured grinding surface. With the advantage of wavelet multiscale decomposition, the decomposed signal was processed.

### 2.2. Surface Topography Decomposition

Wavelet analysis has the advantages of multiscale decomposition and high-resolution interpretation, which was widely used in the field of signal processing [26,30]. Based on the theory of wavelet analysis, the surface topography signals were decomposed multiscale. Each scale of signal can be decomposed into high-frequency and low-frequency components. The process of decomposition was to extract the high-frequency signal from the signal, and then decompose the low-frequency signal to the next scale. With the increase of decomposition scales, multiscale and high-resolution interpretation of complex topography signals can be realized. The wavelet energy method was the method which can identify the main component of the complex signal through analyzing the energy of the decomposed signals [26,28]. The square sum of the data at each decomposition scale was used as the surface topography energy. After the additional component were stripped off, the main component of the topography signal can be extracted by searching for the energy mutation point.

Regardless of other factors, the grinding surface topography was only related to the machining parameters and grinding wheel parameters, that was, the TFB [26]. The TFB was the main component of the grinding surface topography and can be constructed by GSM. In addition to the TFB, both HFB and LFB were additional components. From the perspective of signal decomposition, the wavelet energy mutation phenomenon appears among the HFB, the LFB, and the TFB. The main component of the surface topography (TFB) can be separated by finding the energy mutation point.

According to above process, the wavelet analysis was used to decompose the surface topography in multiple scales, and the wavelet energy method was used to divide the topography signal into three frequency bands: HFB, TFB, and LFB. Figure 2 shows the signal decomposition flow chart of grinding surface topography. Combined with the causes of grinding surface, the following classification was made: the TFB corresponded to the theoretical topography components in the measured topography and was affected by processing parameters and grinding wheel parameters; the component whose frequency was higher than the TFB was attributed to the HFB, which corresponds to the high-frequency topography component in the measured topography, and was affected by the high-frequency vibration in the processing system. The component whose frequency was lower than the TFB was attributed to the LFB, which corresponds to the low-frequency topography component in the measured topography, and was affected by the low-frequency vibration of the processing system, the feed error of grinder, and other factors.

Through above decomposition process, the influencing factors in the grinding process can be classified according to the frequency information, and the analysis transformation from numerous parameters to three parameters can be realized. Compared with that of the analysis of multiple parameters separately, only three frequency bands of the measured topography need to be analyzed, which greatly reduced the difficulty of the analysis process.

### 2.3. Simulation Process of Surface Topography

Based on the decomposition process in Section 2.2, the measured topography can be divided into three components. The separated the HFB and the LFB were extracted, and then reconstructed them to the digital combination model. The digital combination model was applied to the generation of grinding surface under other processing parameters. Furthermore, the extraction method of the HFB and the LFB conformed to ISO25178-3 [31]. The simulation process of grinding surface topography was shown in Figure 3.

The separated TFB corresponded to the theoretical topography component. In addition, the theoretical topography can be constructed using GSM with specific machining parameters and grinding wheel parameters. Eventually, combined with the digital combination model and the theoretical topography, the simulation topography of grinding surface under different processing parameters can be generated.

## 3. Decomposition and Extraction of Surface Topography

The simulation method of surface topography was described in Section 2. The decomposition and extraction process of measured surface topography will be introduced in this section. Because the presented method was based on the analysis of the measured topography, the samples surfaces need to be ground and measured primarily.

### 3.1. Measurement of Surface Topography

The MA7130D/H CNC (Shenyang Machine Tool CO., LTD, Shenyang, Liaoning, China) grinder was used to grind 40Cr material samples. The grinding wheel radius (*r_s_*) was 200 mm and the grain size of grinding wheel was 80#. Meanwhile, the machining parameters of the sample were recorded, which included the speed of grinding wheel *n* = 1460 r/min, the feed speed of the workpiece *v_w_* = 5 m/s, and the grinding feed rate *a_p_* = 5 μm.

After the sample was machined, the noncontact micro topography measurement system ZYGONexView (ZYGO Corporation, Middlefield, CT, USA) was used to measure the surface topography of the grinding sample. The measurement methods were based on the white light scanning and phase-shifting interferometry. The noise-separation algorithms were carefully considered the analyzed detail where dimples [32]. The vertical resolution of the system was 0.1 nm and the RMS repetition accuracy was 0.005 nm, which can realize the high-precision measurement of rough surface. The sampling method complied with ISO25178-3 [31], and the sampling area conformed to the S-F surface. Figure 4 showed the measured topography of the grinding surface under *Sa* 0.403 μm. The sampling area was 3 mm × 3 mm and the number of sampling points was 1024 × 1024.

### 3.2. Wavelet Decomposition of Surface Topography

For the selection of wavelet basis functions in grinding surface decomposition, Zhao compared Haar, dbN, bior, symN, and other wavelet basis functions from the perspective of the overall error of reconstructed topography [26]. When db9 was selected as wavelet basis function, the error was minimum. Therefore, db9 wavelet basis function was selected in this paper. In addition, the selected db9 wavelet basis function was an orthogonal basis wavelet function. Before and after wavelet decomposition, the wavelet energy of surface topography was conserved. For the selection of decomposition scales (*i*), the maximum decomposition scale was calculated according to Equation (1). When *n* was 1,024, the maximum decomposition scale was 9. Wavelet energy was the square sum of the data at each decomposition scale [28], which was calculated according to Equation (2).
(1)$$i=\left[\log_{2}(n)\right]-1$$
(2)$$E_{energy}=\sum_{x=1}^{n}s{\left(x\right)}^{2}$$

According to the method described in Section 2.2, the surface topography signal was decomposed by the db9 wavelet basis function in 9 scales. The high-frequency reconstruction topography under each scale was shown in Figure 5. After wavelet decomposition, the wavelet energy at each decomposition scale can be obtained. Table 1 showed the wavelet energy and its proportions of low-frequency reconstructed topography under different decomposition scales.

As can be seen from Table 1, there were two energy mutation points, from scale 3 to scale 4 and from scale 7 to scale 8, where the wavelet energy decreases greatly. According to these points, the measured topography can be divided into three frequency bands: HFB, TFB, and LFB. Combined with the high-frequency reconstruction topography under different scales in Figure 5, the reconstruction topographies at scales 1, 2, and 3 correspond to the HFB information in the measured topography signal, which was caused by the influence of high-frequency vibration in the processing system. The reconstruction topographies at scales 8 and 9 correspond to the LFB information in the measured topography signal, which was caused by the influence of the low-frequency vibration of the processing system, the feed error of grinder, and other factors.

The reconstructed topographies at scales 1, 2, and 3 were integrated together as the HFB component. Meanwhile, the reconstructed topographies at scale 8 and 9 were integrated as the LFB component. The reconstructed topographies of the HFB and the LFB constituted the digital combination model, as shown in Figure 6.

The digital combination model consists of both HFB and LFB. To verify the accuracy of the digital combination model, the HFB and LFB were extracted separately from the measured topographies under different machining parameters. Before processing of component extraction, the grinding surface topographies under different machining parameters was measured. The feed rate of grinding (a) was taken 1 μm, 2 μm, 5 μm, 10 μm. The workpiece feed rate (*v_w_*) was taken 1 m/s, 5 m/s, 10 m/s, 15 m/s. According to the above process, the reconstructed topographies of the HFB and the LFB were extracted separately. The root mean square deviation (*Sq*) under reconstructed topographies of the HFB and the LFB were compared separately. In addition, three different areas of the machined surface under different machining parameters were displayed.

The comparison results of *Sq* was shown in Figure 7. The *Sq* of the reconstructed topographies under different machining parameters were small. In addition, the maximum mean square error of HFB was 1.59%, and the maximum mean square error of LFB was 2.33%, which verified the accuracy of the digital combination model.

## 4. Simulation of Theoretical Topography

Through the analysis of Section 3, the HFB and the LFB components of the measured topography were extracted and the digital combination model was obtained. The TFB component corresponded to the theoretical topography. Based on the theory of grinding kinematics, GSM was used to construct the theoretical topography in this section. The theoretical topography was related to the grinding wheel parameters and processing parameters. The grinding wheel parameters were described primarily.

### 4.1. Topography Simulation of Grinding Wheel

Generally, the surface topography of grinding wheel can be simulated by non-Gaussian random surface [22]. Therefore, based on FFT and Johnson transform system, the non-Gaussian surface with specific roughness parameters (arithmetic mean deviation *Sa*, deviation from root mean square *Sq*, skewness *Ssk*, kurtosis *Sku*) was generated [11].

The ZYGONexView (ZYGO Corporation, Middlefield, CT, USA) was used to measure the grinding wheel surface under 80# granularity. Four parameters including the arithmetic mean deviation *Sa*, root mean square deviation *Sq*, skewness *Ssk*, and kurtosis *Sku* were selected for comparison. Four parameters were common parameters to characterize rough surfaces [33,34]. The parameters conformed to ISO25178-2 [35]. *Sa* was the basic characterization parameter of rough surface; the other three were central moment parameters. After data processing, the roughness parameters of grinding wheel were obtained as shown in Table 2. To ensure the accuracy of the simulation topography, three areas of the grinding wheel surface were collected for analysis and processing. Finally, the average value of each parameter was taken to simulate the surface topography of the grinding wheel.

Combined with the roughness parameters of grinding wheel surface and the process described in reference [17], the simulation topography of grinding wheel can be realized by using MATLAB software (R2018b). Figure 8 showed the simulation topography of the grinding wheel under 80# granularity, and the simulation area was 3 mm × 3 mm.

### 4.2. Theoretical Topography Simulation of Grinding Surface

For the simulation of the theoretical topography of grinding surface, the implementation method was described in reference [18]. Combined with simulation topography of the grinding wheel in Section 4.1 and the machining parameters in Section 3.1, the theoretical topography of grinding surface can be realized by using MATLAB software. The simulation flow chart of grinding surface topography was shown in Figure 9.

In the process of simulating the grinding surface topography, it was necessary to determine the size of the simulation area and discrete it primarily. After that, the topography data of the grinding wheel and the machining parameters were inputted. Not all abrasive particles were involved during the grinding process, so it was necessary to judge whether the grinding point was effective before proceeding to the next process. The effective grinding point was participated in the cutting during the grinding process. The criterion for judgment was the correlation between the topography data of the grinding wheel and the grinding feed rate. Then, grinding trajectory of abrasive particles were calculated, and the machining path of abrasive particles on workpiece surface were solved. The formation of grinding surface topography was the result of the joint action of all abrasive particles on the workpiece in the grinding process. The minimum value of data points under the action of the particles on each vertical section will form the final grinding surface topography. Figure 10 showed the theoretical topography under *v_w_* = 5 m/s and *a_p_* = 5 μm.

## 5. Results Comparison

According to the simulation process of grinding surface topography (Figure 4), combined with the digital combination model obtained in Section 3.2 and the theoretical topography obtained in Section 4.2, the simulation topography of grinding surface can be constructed. An important issue need to be illustrated: the phase of the topography signal. As can be seen from Figure 6, the digital combination model was closer to the wide amplitude surface, which can be more suitable for the reconstruction of surface topography at medium scale. However, for different spatial scales, the digital combination model shown in this paper was not universal. For the characterization of surface topography at larger or smaller scales, the digital combination model needs to be extracted again. Moreover, for the characterization of smaller surface topography, the phase of topography signal needs to be quantified. The processing method for the phase of topography signals at different scales was mentioned in the reference [36]. Combined with the processing method for the phase and the presented method in this paper, the reconstruction of surface topography at different scales can be realized. Here, only the reconstruction of surface topography at medium scale was shown. The simulation topography under processing parameters *v_w_* = 5 m/s and *a_p_* = 5 μm was shown in Figure 11. Figure 11 was the superposition result of the Figure 10 and the digital combination model. Compared with Figure 10, Figure 11 consists of the HFB component, the LFB component and the theoretical topography component, which was the eventual outcome of simulation topography.

To verify the accuracy of the simulation method, the roughness parameters of simulation topography and measured topography under different machining parameters were compared. Due to limited space, only the simulation topography and measured topography under different feed speeds were selected for comparison. *v_w_* was 1 m/s, 5 m/s, 10 m/s and 15 m/s, respectively, and other machining parameters were the same as described in Section 3.1.

To ensure the reliability of the data, data acquisition was carried out in three areas of the simulation topography and measured topography respectively. In addition, the three surfaces were randomly selected to ensure the universality of the analysis results.

The comparison of roughness parameters between measured topography and simulation topography were shown in Figure 12. Excitingly, the change trend of measured topography and simulation topography was similar, and the values of the standard deviation were insignificant. The relevant data can be found in Appendix A. Furthermore, the relative error of the roughness parameters between measured topography and simulation topography were compared. The comparison results were shown in Appendix A. As shown in Appendix A, the maximum relative error of *Sa*, *Sq*, *Ssk* and *Sku* were 1.79%, 2.24%, 4.69%, and, 4.73%, respectively.

## 6. Conclusions

In summary, a novel simulation method of topography for grinding surface was proposed in this paper. In this method, wavelet analysis was used to extract the digital combination model from the measured topography. Then, combined with the digital combination model and the theoretical topography obtained by GSM, the simulation of grinding surface topographies under different machining parameters were realized. Furthermore, the roughness parameters of simulation topography and measured topography were compared. The comparison results showed that the roughness parameters error was insignificant, which verifies the feasibility and accuracy of the presented method in this paper. This method provides a new way for the simulation of grinding surface topography. However, with the development of signal extraction technology, wavelet analysis may not be the optimum processing method. A signal extraction method more suitable for grinding surface topography needs to be further researched.

## Figures and Tables

**Figure 1 materials-14-05128-f001:**
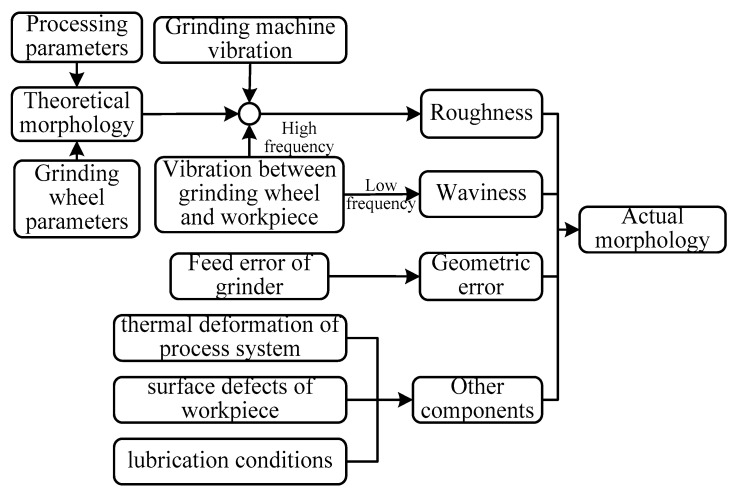
Causes analysis of surface topography in surface grinding.

**Figure 2 materials-14-05128-f002:**
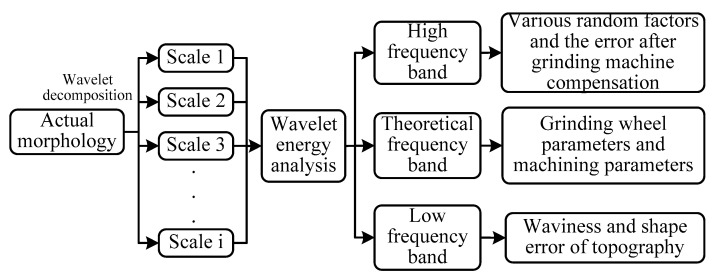
Decomposition process of surface topography signal.

**Figure 3 materials-14-05128-f003:**
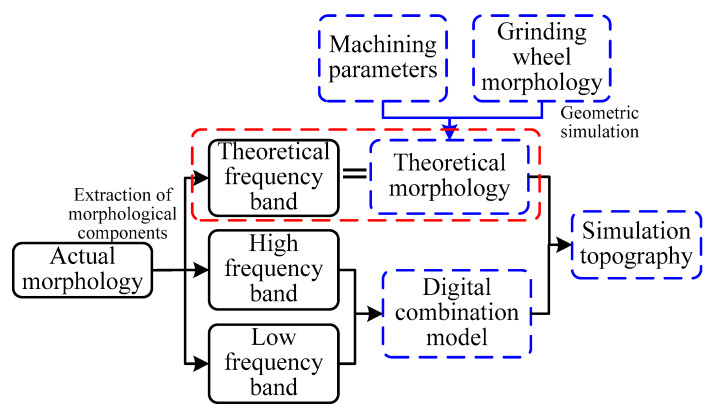
Simulation process of grinding surface topography.

**Figure 4 materials-14-05128-f004:**
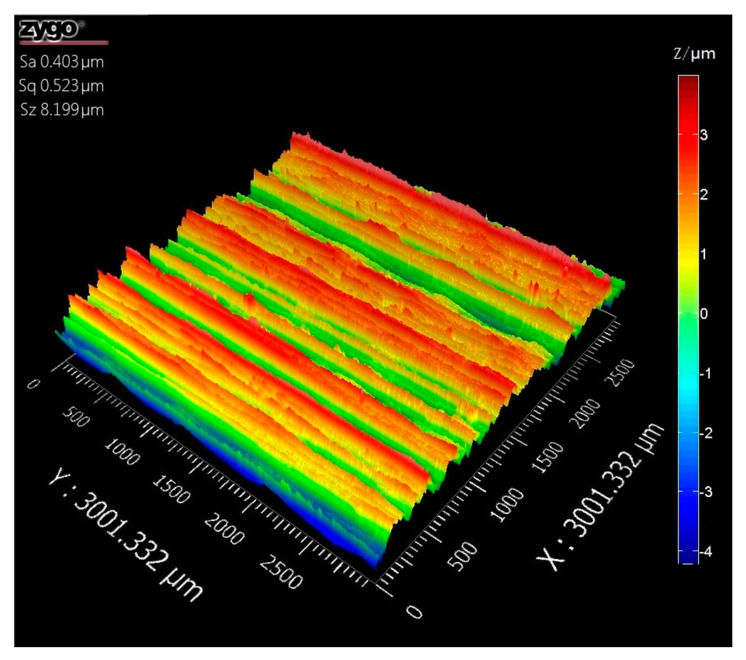
Measured topography of grinding surface under *Sa* 0.403 μm.

**Figure 5 materials-14-05128-f005:**
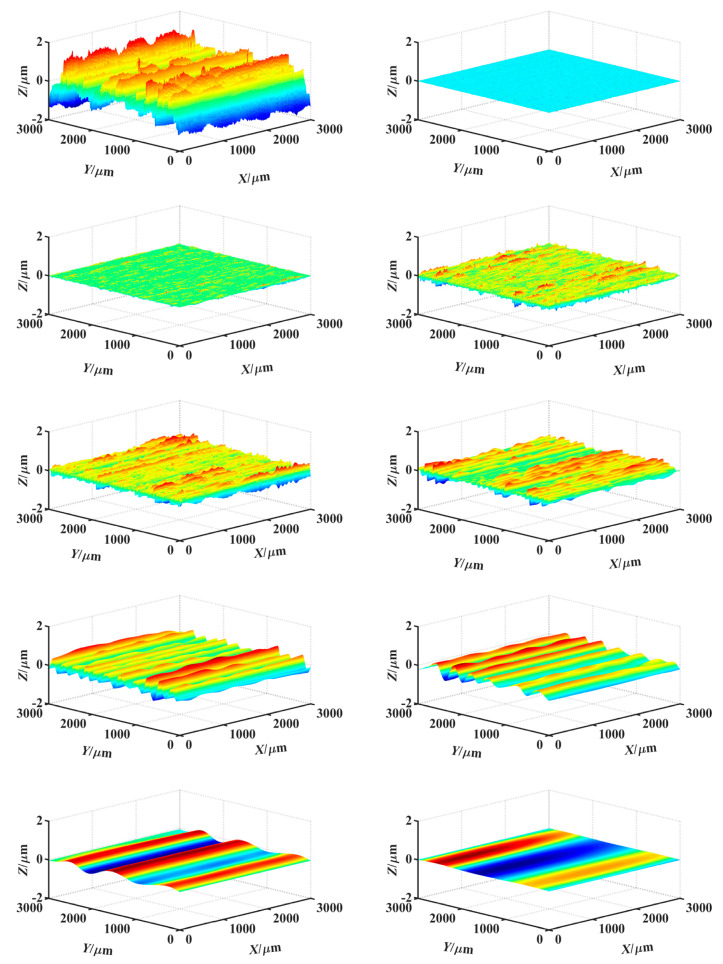
Reconstruction topographies of high-frequency under different decomposition scales.

**Figure 6 materials-14-05128-f006:**
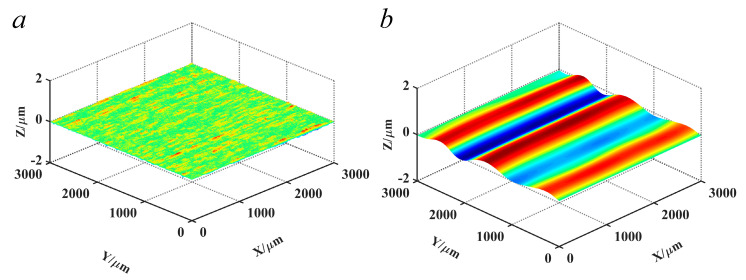
Digital combination model. (**a**) Reconstructed topographies of the HFB; (**b**) reconstructed topographies of LFB.

**Figure 7 materials-14-05128-f007:**
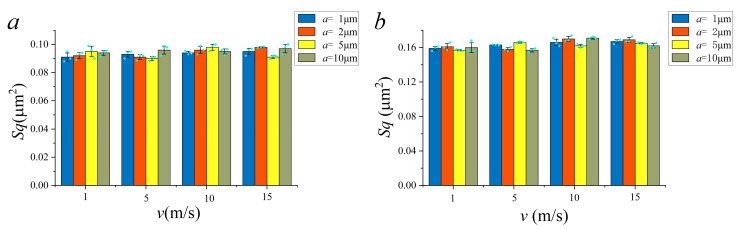
Comparison results of *Sq*. (**a**) *Sq* of HFB; (**b**) *Sq* of LFB.

**Figure 8 materials-14-05128-f008:**
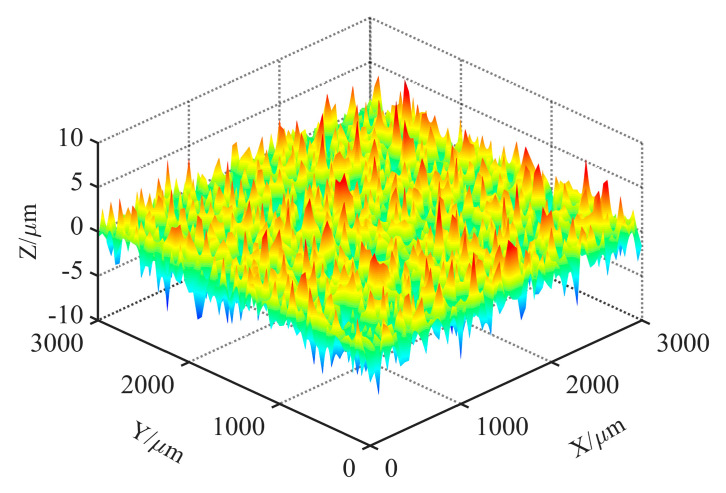
Simulation topography of grinding wheel.

**Figure 9 materials-14-05128-f009:**
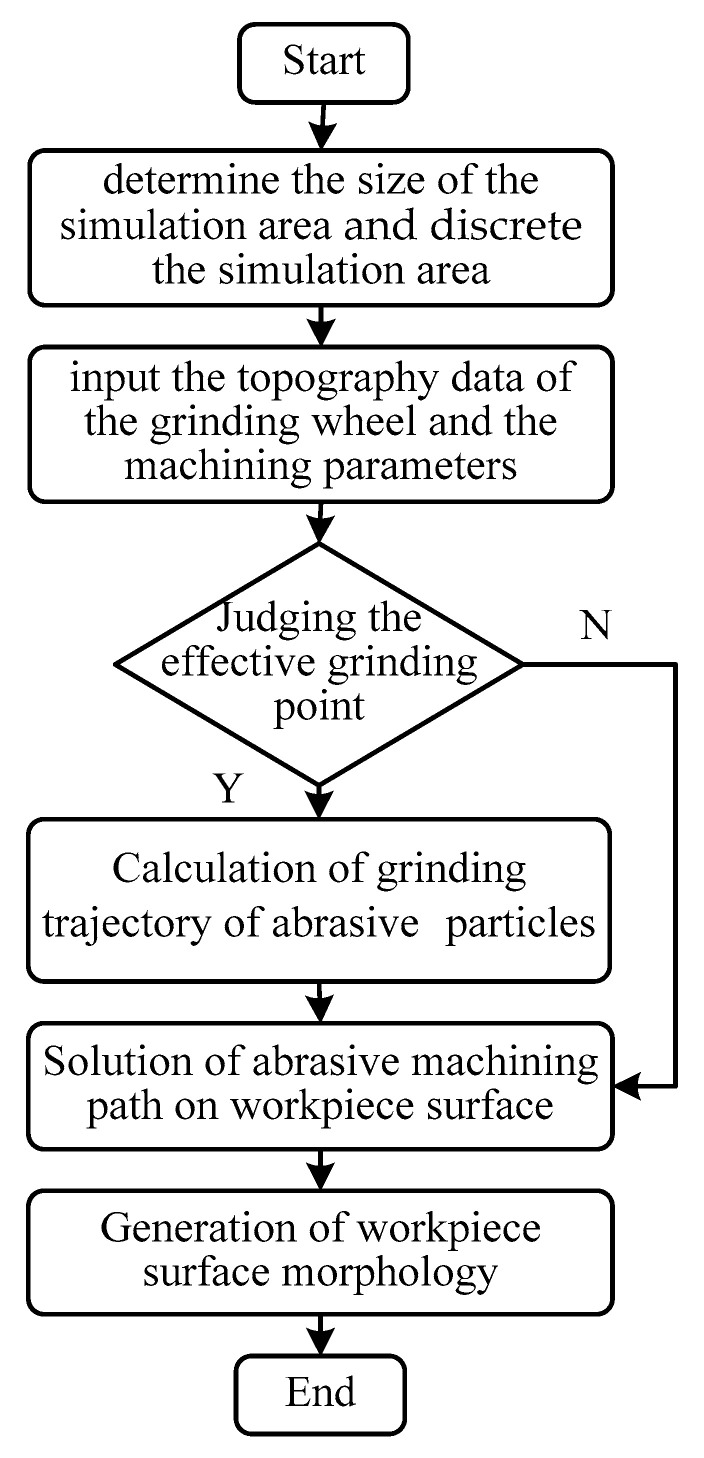
Simulation flow chart of grinding surface topography.

**Figure 10 materials-14-05128-f010:**
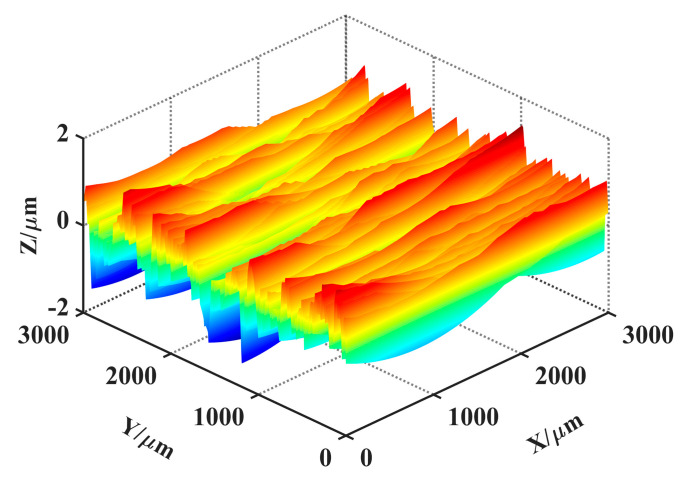
Theoretical topography under *v_w_* = 5 m/s and *a_p_* = 5 μm.

**Figure 11 materials-14-05128-f011:**
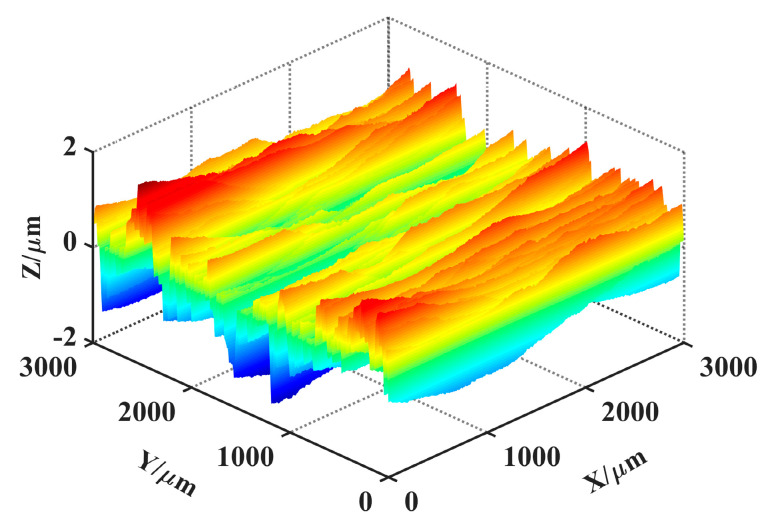
Simulation topography under *v_w_* = 5 m/s and *a_p_* = 5 μm.

**Figure 12 materials-14-05128-f012:**
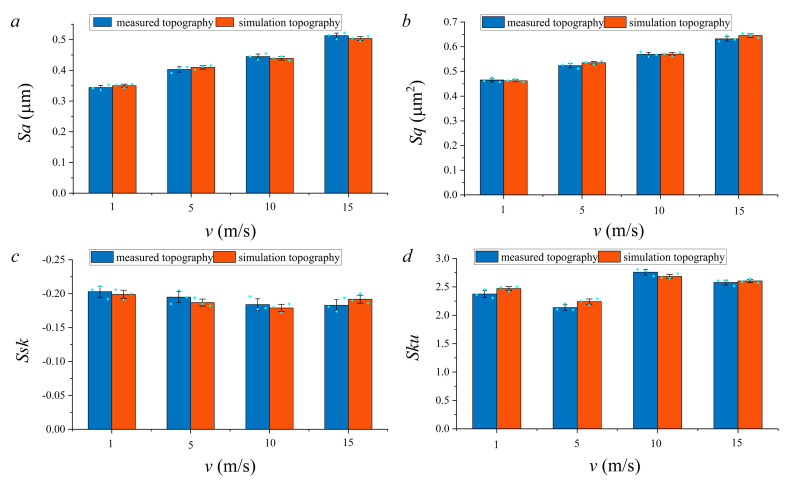
Comparison results of roughness parameters. (**a**) *Sa*; (**b**) *Sq*; (**c**) *Ssk*; (**d**) *Sku*.

**Table 1 materials-14-05128-t001:** Wavelet energy and its proportions of low-frequency reconstructed topography under different decomposition scales.

Original Topography		Decomposition Scale								
		1	2	3	4	5	6	7	8	9
Wavelet energy/(μm^2^)	239,798	239,751	239,057	231,226	211,915	185,456	121,233	77,612	19,232	8113
Proportion/%	100	99.99	99.69	96.43	88.37	77.34	50.56	32.37	8.12	3.38

**Table 2 materials-14-05128-t002:** Roughness parameters of grinding wheels with different areas.

Sampling Area	*Sa*/μm	*Sq*/μm	*Ssk*	*Sku*
1	11.210	12.235	−0.374	3.575
2	11.612	11.975	−0.328	3.867
3	11.321	12.413	−0.354	3.914
Mean value	11.381	12.208	−0.352	3.785

## Data Availability

Not applicable.

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
