# Peer review of "A Novel Simulation Method of Micro-Topography for Grinding Surface"

_materials, 2021, doi:10.3390/ma14185128_

Round 1
Reviewer 1 Report
In lucrare este prezentata o noua metoda of micro-topography for grinding surface. Based on the theory of wavelet analysis, multi-scale decomposition of the measured topography was conducted. The topography was divided into high frequency band (HFB), theoretical frequency band (TFB) and low frequency band (LFB) by wavelet energy method.
Recomand publicarea ei dar doar dupa efectuarea modificarilor solicitate in fisierul atasat.

Author Response
We thank the reviewers for his careful read and thoughtful comments on the previous manuscript. We have carefully taken his comments into consideration in preparing our revision, which has resulted in a paper that is clearer, more compelling, and broader. The following attached file was the response to the reviewer comments.

Reviewer 2 Report
Dear author(s), please find below suggestions that may, at least partially, justify my final evaluation of the reviewed manuscript ‘A novel simulation method of micro-topography for grinding surface’, Manuscript ID: materials-1323741.
Paper can be classified as an encouraging, proposed method is interesting. Nevertheless, there are many, in my feelings, non-completely resolved issues that make the paper complicated to clearly understand by the reader. The presented method can be valuable in the field of surface topography characterisation (modelling) that digital actions completed over the measured data are increasingly important in the analysis of the tribological properties of machined parts. However, many details must be radically improved:
- First of all, how was the accuracy of the HFB, TFB and LFB decomposition verified? Based on the conclusion ‘The multi-scale decomposition of grinding topography was carried out by using 288 wavelet analysis theory.’ (lines 288-289), how can we conclude that the Daubechies wavelet (e.g. Db9) is suitable for a decomposition and definition of the HFB, TFB and LFB components of the surface topography data? Simultaneously, the selection of the degree (db9) of a decomposition process was not also resolved. Therefore, the mentioned statement seems to be far-reaching if a comparison with different methods (e.g. other orthogonal-based wavelet functions with similar decomposition properties) was not provided. For an analysis of surface texture, there were many filters proposed for a decomposition (extraction) of surface topography components. I totally agree that wavelets may be (and are) exceedingly advantageous for the characterisation of surface topography, however, in a considered instance, its applicability according to regular methods (e.g. ISO-preferred various Gaussian filters and their modifications) should be considerably improved. The proposed ‘novel’ method seems to be interesting, nevertheless, the definition (decomposition) of HFB, TFB and LFB components must be more clearly justified. Even methods are crucial, not precisely defined decomposition can cause a huge distortion (disproportion) of a modelled data with respect to those measured.
- From the reference list, in my opinion, it should be also markedly improved that only 1 position (from 21 overall) was published in the last 3 years. The more recent publication should be considered that all received make the reader ‘out-of-date’, but the field of research is topical. In fact, all the references were selected correctly but should be completed. Please find below some suggestions for surface topography simulation, which can make the introduction section more up-to-date:
- Lipiński, D., Bałasz, B. & Rypina, Ł. Modelling of surface roughness and grinding forces using artificial neural networks with assessment of the ability to data generalisation. Int J Adv Manuf Technol 94, 1335–1347 (2018). https://doi.org/10.1007/s00170-017-0949-y
- Pawlus P., Reizer R., Wieczorowski M. A review of methods of random surface topography modeling. Tribology International 2020, 152, 106530. https://doi.org/10.1016/j.triboint.2020.106530
- Zhenzhong Zhang, Peng Yao, Jun Wang, Chuanzhen Huang, Hongtao Zhu, Hanlian Liu, Bin Zou. Nanomechanical characterization of RB-SiC ceramics based on nanoindentation and modelling of the ground surface roughness. Ceramics International 2020, 46(5), 6243-6253. https://doi.org/10.1016/j.ceramint.2019.11.094
- Yuhang Pan, Ping Zhou, Ying Yan, Anupam Agrawal, Yonghao Wang, Dongming Guo, Saurav Goel. New insights into the methods for predicting ground surface roughness in the age of digitalisation. Precision Engineering 2021, 67, 393-418. https://doi.org/10.1016/j.precisioneng.2020.11.001
and other publications:
- Reizer R., Galda L., Dzierwa A., Pawlus P. Simulation of textured surface topography during a low wear process. Tribology International, 2011, 44(11), 1309-1319. https://doi.org/10.1016/j.triboint.2010.05.006
- Reizer R. Simulation of 3D Gaussian surface topography. Wear 2011, 271(3–4), 539-543. https://doi.org/10.1016/j.wear.2010.04.009
- Chakrabarti, Suryarghya.; Paul, S. Numerical modelling of surface topography in superabrasive grinding. The International journal of advanced manufacturing technology, 2008, 39.1-2: 29-38. http://dx.doi.org/10.1007%2Fs00170-007-1201-y
- Further considering the introduction section, there is more state-of-the-art than a critical review of the recent results on the simulated surface topography data. Some critical comments should be found.
- In lines 32-36 there is no explanation why Gauss and non-Gauss surfaces are simulated accordingly to the numerical simulation method (NSM). It makes ordinary reader, not understanding some of the further studies, provided in the next section of the manuscript.
- In paragraphs 62-70, at this stage of the paper, there are no results (references) providing assumptions made. The proposal would be more appropriate than the conclusion (?). Authors present more final conclusions than assumptions based on the critical review of the literature. At this stage of the introduction, some conclusion(s) based on the review of the references should be defined. It feels like a little mess and makes the reader not interesting in further studies when the conclusion is presented in an introduction section.
- More about the introduction, more valuable papers in the Materials journal can be found considering an analysis of surface topography, its measurement results, measurement errors (noise), characterisation and, respectively, data analysis or modelling in general. Some of them should be considered. Please find some examples below:
- Wang, Y.; Mu, X.; Yue, C.; Sun, W.; Liu, C.; Sun, Q. A High Precision Modeling Technology of Material Surface Microtopography and Its Influence on Interface Mechanical Properties. Materials 2021, 14, 2914. https://doi.org/10.3390/ma14112914
- Gogolewski, D.; Bartkowiak, T.; Kozior, T.; Zmarzły, P. Multiscale Analysis of Surface Texture Quality of Models Manufactured by Laser Powder-Bed Fusion Technology and Machining from 316L Steel. Materials 2021, 14, 2794. https://doi.org/10.3390/ma14112794
- Królczyk, G.; Kacalak, W.; Wieczorowski, M. 3D Parametric and Nonparametric Description of Surface Topography in Manufacturing Processes. Materials 2021, 14, 1987. https://doi.org/10.3390/ma14081987
- Kubo, A.; Teti, R.; Ullah, A.S.; Iwadate, K.; Segreto, T. Determining Surface Topography of a Dressed Grinding Wheel Using Bio-Inspired DNA-Based Computing. Materials 2021, 14, 1899. https://doi.org/10.3390/ma14081899
- Khan, A.M.; Jamil, M.; Mia, M.; Pimenov, D.Y.; Gasiyarov, V.R.; Gupta, M.K.; He, N. Multi-Objective Optimization for Grinding of AISI D2 Steel with Al2O3 Wheel under MQL. Materials 2018, 11, 2269. https://doi.org/10.3390/ma11112269
- Ullah, A.S.; Caggiano, A.; Kubo, A.; Chowdhury, M.A.K. Elucidating Grinding Mechanism by Theoretical and Experimental Investigations. Materials 2018, 11, 274. https://doi.org/10.3390/ma11020274
- Calvimontes, A.; Mauermann, M.; Bellmann, C. Topographical Anisotropy and Wetting of Ground Stainless Steel Surfaces. Materials 2012, 5, 2773-2787. https://doi.org/10.3390/ma5122773
- Considering the first comment, how can be multi-scale and wavelet energy method be classified as this correct? It should be clarified for a usual reader that there was no comparison with other methods or, consequently, no references were provided confirming this conclusion.
- The ‘other random factors’ (lines 78-79) should be also defined.
- Another, not fully explained concept is ‘the surface topography energy’ (lines 103-104). For a reader with non-wavelet studies experience, it would make him confused.
- According to the sentence ‘Through above decomposition process, the influencing factors in the grinding process can be classified according to the frequency information. ‘ (lines 129-130), where the frequency information was significant in the whole decomposition process, why an analysis of the Power Spectral Density (PSD) graphs was not proposed? In fact, there is not encouraging response if the received HFB components were extracted correctly. Please find some, only selected, PSD characterisations of the surface topography data in the following papers:
- M. Krolczyk, R.W. Maruda, P. Nieslony, M. Wieczorowski. Surface morphology analysis of Duplex Stainless Steel (DSS) in Clean Production using the Power Spectral Density. Measurement 2016, 94, 464-470. https://doi.org/10.1016/j.measurement.2016.08.023
- Tevis D B Jacobs et al 2017 Surf. Topogr.: Metrol. Prop. 5 013001. https://doi.org/10.1088/2051-672X/aa51f8
- In line 162 there is a sentence ‘3x3 mm2’. It is my understanding that the sampling area was 3mm x 3mm?
- The scales in Figures, e.g. Figure 5, should be unified, especially for the same frequencies, Scales 1, 2 and 3 for HFB, Scales 4-7 for TFB and, consequently, 8 and 9 for LFB.
- In Figure 6, reconstruction of TFB should be also presented to show all of the surface components. When selected frequencies (components) of surface topography are extracted, all of the parts must be defined and presented.
- The digital combination model seems to be a sum of frequency-defined components. Is there any definition of this term? Maybe in some standards (e.g. ISO) or in the literature? There was not clearly presented if that is a novelty, therefore the reader may feel more disoriented, respectively. In ISO standards (ISO 25178-3:2012 in particular) there were defined some operators, e.g. F-operator, S-F surface, S-operator (S-filter); in my opinion, it might be presented according to those standards.
- The selection of parameters (Sa, Sq, Ssk, Sku) for analysis should be also justified and explained. For a usual reader, not affected by the surface topography measurements, it can be complicated to understand the importance of the parameters (lines 269-270).
- According to comment no.15 and the sentence of a ‘data processing’ (line 219), what type of data processing(s) were provided? Are those data processing techniques defined in some standards (ISO or other)? Maybe in some of the practical guides? Included in selected research items (papers)?
- It was not defined how were the three areas (line 221) selected? Randomly? Based on the surface features/characterisation? A description in lines 270-272 is not completed.
- There is no information on measurement noise (errors) and measurement uncertainty.
- Can be differences between Figures 10 and 11 be clarified? If not, the reader can feel misinformed. They seem to be similar except for the changes in X and Y directions. If not, it should be presented and figures directions unified, like scales, mentioned previously.
- Values of the parameters, presented in Figure 12, were not presented except on the graph. Looks like the value of a skewness (Ssk) parameter was stabilised when vw was equal to 10 m/s or greater but, according to the simulation, it was not. It must be fully clarified.
- As in comment no. 20, how can we conclude that differences in values of analysed parameters were small if their values are not given. In some cases, for selected surface topography parameters, variations around 5% of values may be significant (statement of ‘insignificance’, line 299, may be too bold). Without exact values, this found (accuracy) is lost.
From all of the above plenty of suggestions, feel sorry, but found it unsuitable for publication, at least in the current form, in the Materials journal and decided to reject the manuscript. Paper should be radically improved before taking it into resubmission in further processing.
Author Response

(The authors gave the same response as above.)

Round 2
Reviewer 2 Report
Dear author(s), thank you for your responses. Many comments were improved significantly that makes the paper more suitable for considering to publish in the Materials journal. Nevertheless, there are some comments that, in my feeling, were not explained and corrected appropriately:
Comment 14:
‘F-operator, S-F surface, S-operator (S-filter)’ should be presented and correlated (related), if possible, to the LFB and HFB and their ‘combination’. Please find some clarifications in ISO standard:
- ISO 25178-3:2012 Geometrical product specifications (GPS) – Surface texture: Areal – Part 3: Specification operators.
Comment 16: Removing sentence ‘After data processing…‘ did not provide required information about surface topography data processing. There is still no appropriate response if (and how) the data were ‘processed’. Usually, there is no ‘raw measured data’ processed in further digital actions but, before studies, some actions are made. If ‘raw measured data’ were analysed it must be provided.
Comment 18:
Specification of measurement instrument and sentence ‘realize the high-precision measurement of rough surface.’, in my feeling, may be not convincing. Please precise that information. In below there is some suggestion how to define a measurement noise:
- Claudiu L Giusca et al 2012 Meas. Sci. Technol. 23 035008. https://doi.org/10.1088/0957-0233/23/3/035008
- Podulka, P. Reduction of Influence of the High-Frequency Noise on the Results of Surface Topography Measurements. Materials 2021, 14, 333. https://doi.org/10.3390/ma14020333
- Ángela Rodríguez-Sánchez, Adam Thompson, Lars Körner, Nick Brierley, Richard Leach. Review of the influence of noise in X-ray computed tomography measurement uncertainty. Precision Engineering 2020, 66, 382-391. https://doi.org/10.1016/j.precisioneng.2020.08.004
Please try to precisely answer the above issues along with their confirmation in the manuscript.
Author Response
We thank for your careful read and thoughtful comments on the previous manuscript. We have carefully taken his comments into consideration in preparing our revision, which has resulted in a paper that is clearer, more compelling, and broader. The following summarizes how we responded to the reviewer comments.
